# Elucidating the complex membrane binding of a protein with multiple anchoring domains using extHMMM

**Jesper J. Madsen** [1,2]*, **Y. Zenmei Ohkubo** [3]*

**1** Department of Molecular Medicine, Morsani College of Medicine, University of South Florida, Tampa, Florida, United States of America, **2** Center for Global Health and Infectious Diseases Research, Global and Planetary Health, College of Public Health, University of South Florida, Tampa, Florida, United States of America, **3** Department of Bioinformatics, School of Life and Natural Sciences, Abdullah Gül University, Kayseri, Turkey

* jespermadsen@usf.edu (JJM); yzohkubo@gmail.com (YZO)

## Abstract

Membrane binding is a crucial mechanism for many proteins, but understanding the specific interactions between proteins and membranes remains a challenging endeavor. Coagulation factor Va (FVa) is a large protein whose membrane interactions are complicated due to the presence of multiple anchoring domains that individually can bind to lipid membranes. Using molecular dynamics simulations, we investigate the membrane binding of FVa and identify the key mechanisms that govern its interaction with membranes. Our results reveal that FVa can either adopt an upright or a tilted molecular orientation upon membrane binding. We further find that the domain organization of FVa deviates (sometimes significantly) from its crystallographic reference structure, and that the molecular orientation of the protein matches with domain reorganization to align the C2 domain toward its favored membrane-normal orientation. We identify specific amino acid residues that exhibit contact preference with phosphatidylserine lipids over phosphatidylcholine lipids, and we observe that mostly electrostatic effects contribute to this preference. The observed lipid-binding process and characteristics, specific to FVa or common among other membrane proteins, in concert with domain reorganization and molecular tilt, elucidate the complex membrane binding dynamics of FVa and provide important insights into the molecular mechanisms of protein-membrane interactions. An updated version of the HMMM model, termed extHMMM, is successfully employed for efficiently observing membrane bindings of systems containing the whole FVa molecule.

## Author summary

Understanding the intricacies of protein-membrane interaction is essential for fleshing out the functional roles of membrane proteins. Substantial evidence indicates that the binding of some proteins, such as coagulation factor Va, proceeds in a multi-step manner requiring some form of rearrangements within the structure and dynamics of the protein,

**Data Availability Statement:** Simulation trajectory datasets are available at Zenodo (references 97-100: doi: 10.5281/zenodo.7813482, doi: 10.5281/zenodo.7813772, doi: 10.5281/zenodo.7813913, and doi: 10.5281/zenodo.7814051). Scripts used

for analysis are available at GitHub (https://github.com/jjmadsen/pub-fva-hmmm.git)".

**Funding:** The author(s) received no specific funding for this work.

**Competing interests:** 'The authors have declared that no competing interests exist.

as well as in the protein's orientation relative to the membrane surface. In the case of factor Va, the intricate binding process can be attributed to the presence of multiple membrane-anchoring domains. In order to feasibly study the dynamics of these mechanisms, we employ an enhanced membrane-mimetic model, HMMM, capable of capturing such processes and rearrangements and making new discoveries possible using the molecular dynamics technique. The successful execution of these studies demanded several modifications to the original implementation of the HMMM. Our exploration not only improves our understanding of FVa's membrane binding dynamics but also contributes to the broader molecular mechanisms governing protein-membrane interactions.

## Introduction

It is necessary for some proteins to bind membrane surfaces because this interaction facilitates important roles in cellular processes such as signal transduction [1–3], adhesion [4–6], and regulation of membrane permeability [7–10]. Proteins can bind to cellular membranes through various mechanisms [11,12]. The driving forces for such interactions are typically non-covalent, including hydrophobic, ionic, and hydrogen bond interactions between the chemical moieties of the protein and the membrane lipids [12]. Any given protein will usually employ a delicate balance between the three types of non-covalent interactions to achieve functional adsorption to the membrane surface, be it generic or specific. The membrane binding process gets much more complex when the protein contains multiple anchoring domains. Such proteins can bind to the membrane surface in qualitatively different ways [13], depending on the nature and number of these domains. The binding can happen in a cooperated manner, independently, or in a hierarchical fashion and the precise process can have important implications for the functions of the protein, e.g., its localization and activity.

Coagulation factor Va (FVa; "a" for activated) is a large glycoprotein that plays a critical role in the blood coagulation cascade [14]. The precursor FV is synthesized as a single-chain protein in the liver and is present in circulation, where it can be activated by proteolysis at three sites by thrombin or factor Xa, releasing its B domain [15]. The proper membrane binding of FVa is crucial in the intersection of the intrinsic and extrinsic pathways, where facilitates the conversion of prothrombin into thrombin, which in turn converts fibrinogen to fibrin. Structural analysis of FVa [16–20] and the prothrombinase complex [21,22] have provided unique insights into prothrombinase function, but the explicit structural details of membrane binding has remained elusive. It is known that FVa binds to the membrane surface via a combination of generic and specific interactions between its light chain and phospholipids in the bilayer [23]. The FVa light chain contains dual "C2-like" discoidin domains, C1 and C2, which are the anchoring domains responsible for phosphatidylserine (PS)-binding and they are essential for cofactor activity [24,25]. Binding of FVa to anionic phospholipids is thought to be driven by electrostatic interactions between certain positively charged amino acids of the protein and negatively charged headgroups of the phospholipids [25,26]. In addition, the membrane anchoring loops of the discoidin domains, the "spikes," contain several hydrophobic amino acid residues that are believed to insert into the membrane upon binding [27,28]. However, despite extensive research conducted in the membrane binding of FVa, the molecular details of the FVa-phospholipid interaction are not well understood. This includes specific amino acid residues as well as the dynamic conformation of FVa before, during and after binding. Progress in this area holds promise for development of new therapeutic strategies, e.g., for

the treatment of thrombotic disorders by regulating membrane binding of FVa in order to control the enzymatic activity of prothrombinase, of which FVa serves as cofactor.

We investigate the process of the membrane binding of FVa to bilayer lipid membranes at unparalleled spatiotemporal resolution by performing extensive molecular dynamics simulations using the most realistic available model structure [29] containing complexed ions and post-translational modifications that was employed for molecular dynamics of the binary complex with prothrombin in aqueous solution, and not for membrane binding as in the present work. The free FVa in solution was briefly studied by molecular dynamics, as well [30]. A series of independent binding events are observed by virtue of the highly mobile membrane-mimetic (HMMM) model, [31] a proven methodology for circumventing the usual obstacles limiting such investigations [31–38]. An updated version of HMMM in terms of both lipid topology and restraints is proposed for the successful application to a system size needed to contain the whole FVa molecule, eliminating the problems of lipid popping-out and membrane undulations.

## Results

### Mode of FVa membrane binding

The process of binding FVa to the membrane proceeds by the interaction of specific amino acid sidechains with moieties of the membrane lipids. Typically, initial contact and insertion of the C2 and C1 domain spikes progress sequentially, although there is variation among the different simulated trajectories. Also recall that the initial positioning of FVa guides C1 and C2 toward the membrane, consistent with the experimentally established framework. Initially, the C2 domain establishes contacts by means of the basic residue K2114, which interacts with the carboxy group of a PS headgroup. Subsequently, contact is made between K2060/K2087 and another PS headgroup's carboxy moiety. This is followed by interactions involving R2080 and K2157 with yet another PS headgroup. These initial contacts occur within t = 5–10 ns. Hydrophobic insertion occurs after the basic contacts with W2063 and W2064 of C2 spike 1, followed by L2116 of C2 spike 3 at around t = ~10 ns. C1 follows a similar pattern, albeit with a slight delay compared to C2. This likely both is related to the reduced presentation of basic residues on the lower part of C1 compared to C2, and to the truncated spike 1. Following the establishment of C2 contacts (both basic and hydrophobic), C1 initiates contact between K1958/K1954 and R1907 with the carboxy moiety of PS headgroups. Subsequently, contact is made with R2023, R1910, and/or R2027 and PS headgroups. This initial series of basic contacts occurs at t = ~10–20 ns. Following this, hydrophobic insertion takes place with L1957 of C1 spike 3. The insertion of the other spikes in C1 and C2 requires more time and does not reach the same depth of insertion, as will be discussed in detail later. The insertion depths of the spikes and the basic residue–PS interaction patterns keep changing to some degree, but they are not irreversibly dislodged. The established binding seems equivalent to that suggested in the "Martinsried-1999" model [27] based on the X-tal structure of FV-C2, the "Milano-2006" model [28] based on FV-C2 bound to PE membrane, the "Rochester-2013" model [39] with a significantly slanted FVIIIa measured by FRET, and the "Måløv-2015" model [33] from molecular dynamics simulation (we adopt the model notation of our recent review article, Ref. [36]). The predicted hydrogen-bond interactions inside the major PS-binding site of the "Martinsried-1999" model [27] (Fig 5A in Ref. [27]) consisting of K2060, Q2085, and S2115 are not consistently observed in our simulations. Furthermore, deep insertion equivalent to that described as reported with binding to lipid-decorated nanotubes (e.g., "Covently-2008" model [40,41] of FVa and "Galveston-2013" model [42] of FVIIIa) is not observed here. Our extHMMM bears dioctanoyl tails, which are long enough to replicate FVa-lipid interactions in

case the deeply buried FVa is a real, stable membrane-bound form to normal cells. We speculate therefore that these modes of binding are only possible when binding to the lipid nanotube, and that the deeply buried forms are probably different from the way these proteins bind to the membrane surface in nature. The radii of curvature of the nanotubes employed (~1.5 nm and ~8.1 nm) [41,42], makes them substantially smaller and more curved when compared with the most of the activated platelet surface (whose rounded dimension is ~2–3 μm with blebs on the scale of an order of magnitude smaller than that, which is still thicker than the nanotubes employed) [43].

## The dual discoidin domains of FVa facilitates a tilted molecular orientation upon membrane binding

Coagulation factor Va (FVa) binds to the phospholipid membrane when oriented with C1 and C2 domains facing the membrane surface with the initial minimal distance of about 1 nm, at which only a few atoms of the domains have near-zero electrostatic energies with membrane lipids and the vast majority of FVa atoms have no interaction with the membrane (as depicted in Fig 1). Attractive electrostatic interactions cause FVa to move closer to the membrane, with both C1 and C2 domains contacting the membrane surface. After initial contact, hydrophobic residues of the spikes insert into the membrane to consolidate a favorable hydrophobic interaction between FVa and the membrane. This is in alignment with analysis of the chemical and electrostatic properties of the FVa molecular surface, where the linger and hydrophobic section of C2's spike 1 protrudes from the domain (S1 Fig). Furthermore, the two discoidin domains have extended patches of positively charged residues forming a "charged belt." Specifically, C2 displays this feature on both the front and back sides, while C1 predominantly displays it on the front size (S1 Fig, bottom panels). There appears to be little difference in the binding process between C1 and C2 domains when looking on the time series for the specific $z$-separation, or the distance between the center-of-mass of individual domains of FVa (shown as five colored spheres in Fig 1) and the membrane center plane ($z = 0$; indicated as a dotted horizontal line in Fig 1). Two different settings of the membrane are explored, one with 110% relative area per lipid (rAPL) and the other with 120% rAPL (Fig 2A and 2B). The reference values of 68.3 and 60.4 Å$^2$ / lipid for POPC and POPS were used [44–46]. Importantly, FVa changes its orientation and tilts on the membrane surface once bound (Fig 2C and 2D), with the dual discoidin domains playing a significant role in directing this tilt. Snapshots from individual simulations show how the varying molecular orientation (in order of increasing tilt angle) is positioned relative to the membrane surface (Fig 2E). The dynamic interactions between FVa and the membrane surface over time and the tilt angle of the protein determine what surface regions of the protein that can interact with the phospholipids directly. In addition to C1- and C2-facilitated interactions with the membrane, we transiently observe contact also with extruding loops of A3 but only in the most tilted orientations (as occurring in Memb #1).

## Domain organization of FVa significantly deviates from the crystallographic reference structure both when membrane-bound state and in solution state, with deviation being more pronounced in the solution state

We characterize the structural deviation of FVa from its crystal structure reference, employing the φ (C1-C2 "scissors") and the θ (C2 out-of-plane "kick") angles as metrics (Fig 3A, insert; the precise atomic definitions are listed in the Methods section). Our simulation results indicate that both the membrane-bound (Fig 3A) and solution-state (Fig 3B) ensembles of FVa exhibit significant deviation from the crystal structure reference (labeled "Xtal"). Notably, the

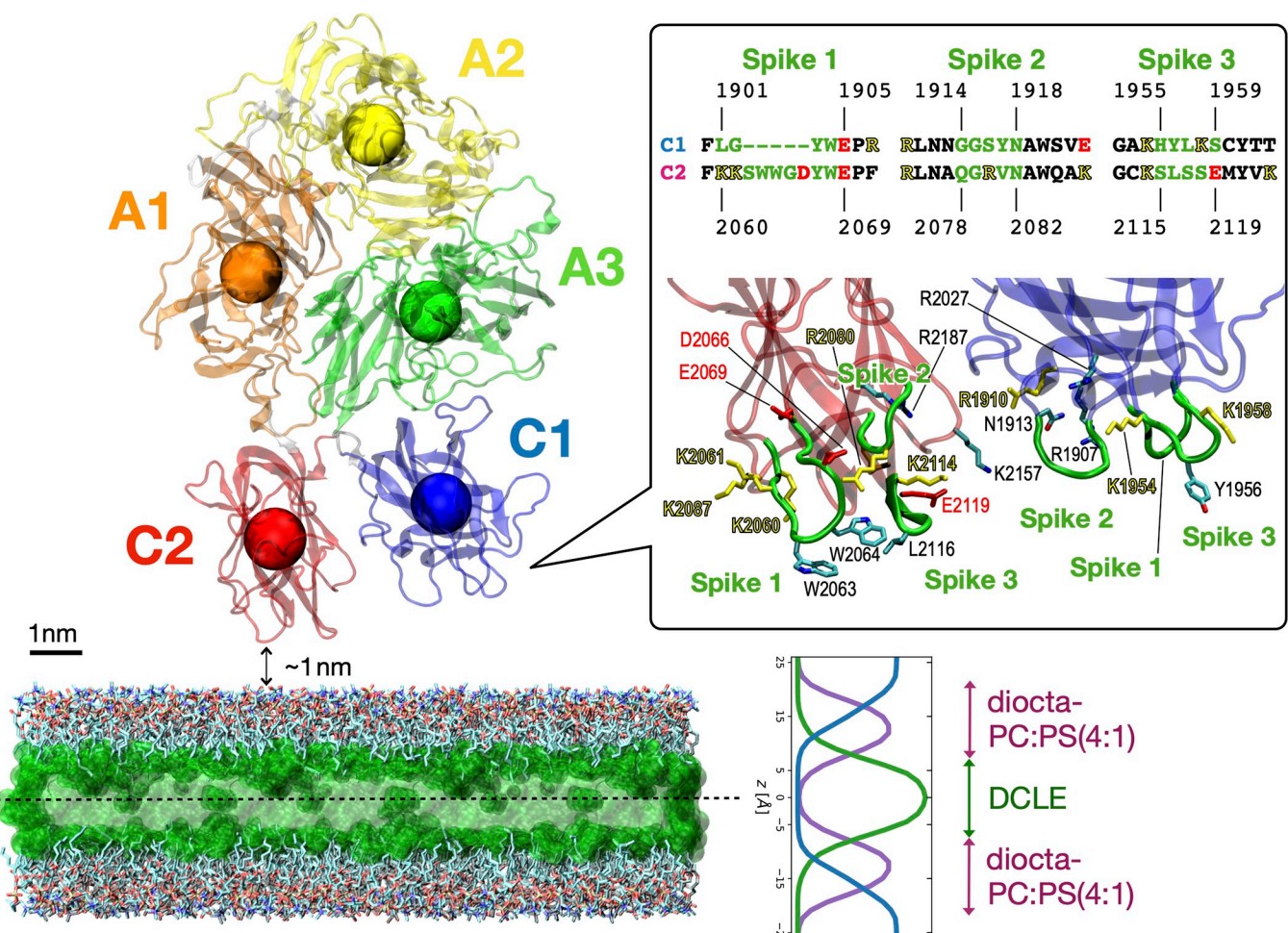

**Fig 1. Initial setup of the FVa protein with respect to the membrane surface.** FVa is positioned facing the membrane with its two membrane-binding domains, C1 and C2. The Fig is a ribbon diagram of FVa with its individual domains colored (heavy chain: A1 in orange and A2 in yellow; light chain: A3 in green, C1 in blue, and C2 in red), and centers-of-mass are indicated by spheres. The membrane consists of a layer of DCLE molecules in the middle (in translucent green) surrounded by explicit medium-length lipids with a headgroup stoichiometric ratio of PC:PS equaling 4:1 in both leaflets. Bulk water molecules (above and below the membrane) are not shown. The inset provides an overview of the sequences of spikes 1–3 regions for both C1 and C2 domains. A zoomed-in view of these regions from the initial setup is shown below, using the same coloring. Certain sidechains are shown in a licorice representation, accompanied by labels.

sampled ranges for the φ and θ angles are approximately [75,135] and [–90,50] degrees, respectively, in the solution state, revealing pronounced discrepancies from the crystal structure. The second and third most populated states in solution ("s1" and "s2") have not equivalent once membrane bound, suggesting a conformational selection process in membrane binding. In addition, the membrane-bound case shows a state ("m1") that is similar, but not identical, to the crystal structure. Two states ("m2" and "s3") are observed with increased φ angle around 100 degrees, indicating some departure from the crystal structure's organization that happens both in solution and when membrane-bound (Fig 3C).

Our analysis has uncovered a significant distinction between the membrane-bound and solution states, particularly regarding the θ angle. This angle exhibits substantial variability in the solution state, leading to the formation of two distinct states ("s1" and "s2") with noticeable rearrangement of the C2 domain against the C1 domain in the "kick" direction, both front and back (see Fig 3B and 3C). Despite these variabilities, our results are consistent with the notion

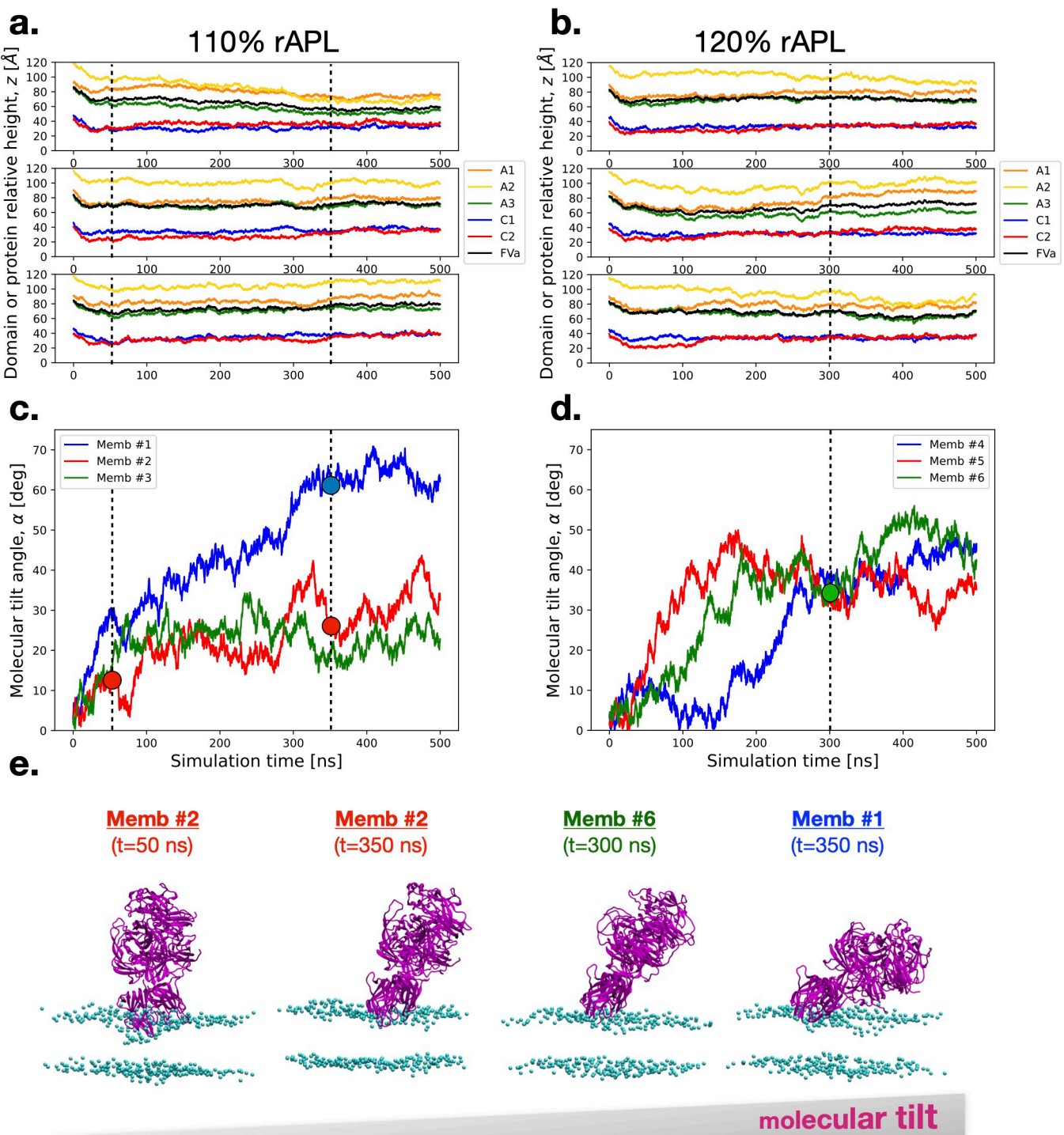

**Fig 2. Spontaneous binding of FVa to the membrane.** Top and middle panels depict time series for specific domain $z$-separation and molecular orientation of FVa, respectively, in two different extHMMM settings: relative area per lipid (rAPL) of 110% (panels **a** and **c**) and 120% (panels **b** and **d**). Panel **e** shows snapshots from the simulations at four different timepoints, each indicated by a dot and dashed lines in panels **a**-**d**. These snapshots depict FVa (ribbon diagram in purple) and the P atoms of the membrane (balls in purple), with the timepoints showing Memb #2 (t = 50 ns), Memb #2 (t = 350 ns), Memb #6 (t = 300 ns), and Memb #1 (t = 350 ns) in increasing order of molecular tilt.

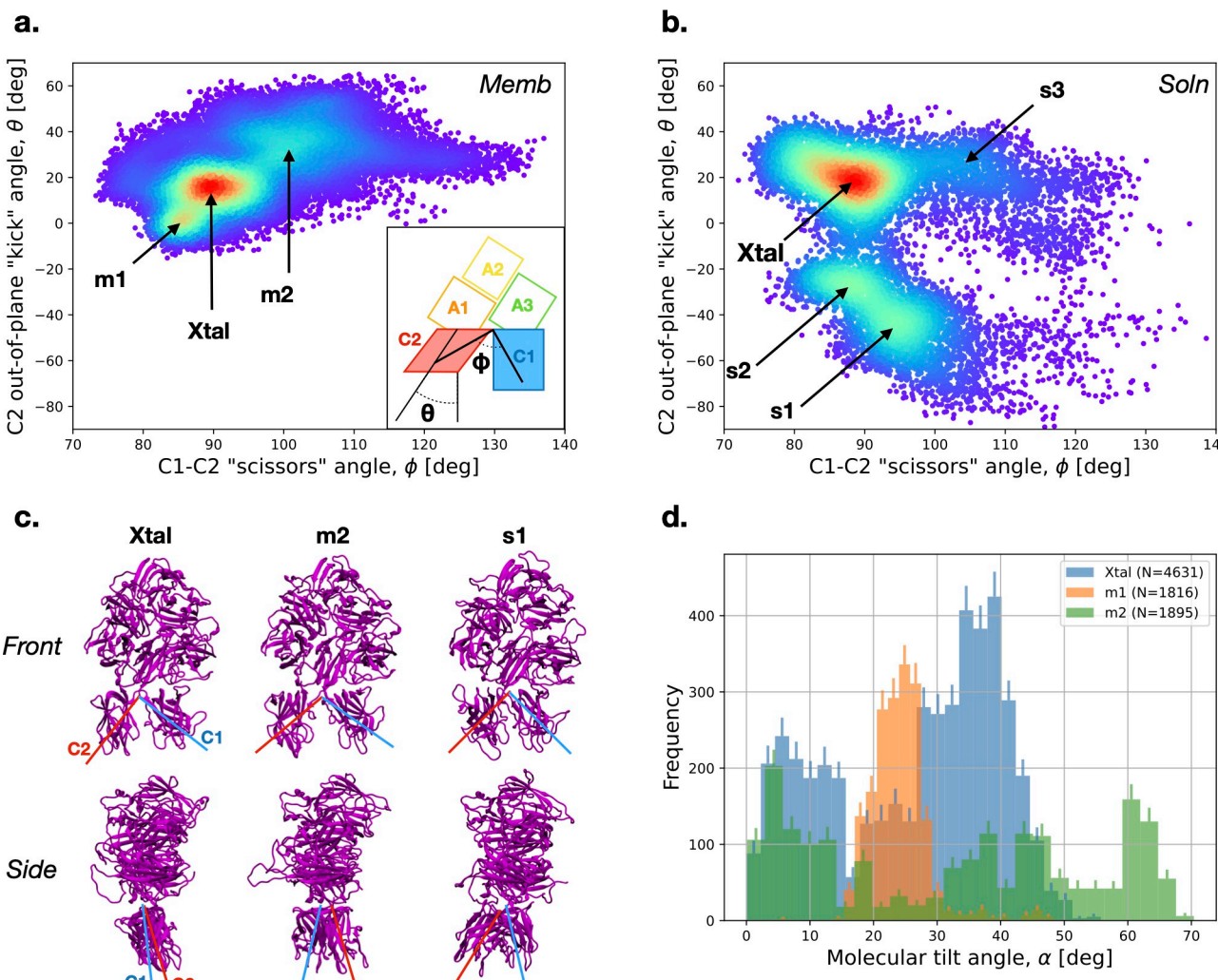

**Fig 3. Domain organization of FVa in solution and when bound to the membrane, as seen in simulations. a.** The "scissors" and out-of-plane "kick" angles of C1-C2 domains in the membrane-bound portion (t = 100–500 ns) of Memb #1–6 trajectories. Inset depicts the definitions of the two angles in a simple schematic. **b.** The "scissors" and out-of-plane "kick" angles of C1-C2 domains in the solution state, as observed in the Soln #1–3 trajectories. **c.** Snapshots from the simulations showing the FVa domain orientation in the Xtal, m2, and s1 sampled basins. The structures are taken from Ref. [29] (Xtal), Memb #1 at t = 300 ns (m2), and Soln #3 at t = 200 ns (s1). The top panels show a front view (same as in Fig 1), while the bottom panels display a side view rotated 90 degrees counterclockwise (when looking down from top) around the protein's first principal axis so that C2 domain is shown in front of C1 domain. The red and blue line segments are positioned to approximately correspond to the "scissors" (top) and "kick" (bottom) angles. **d.** Histogram of the molecular tilt angle of FVa relative to the membrane surface in the Xtal (in blue), m1 (in orange), and m2 (in green) basins. The error bars (90% confidence intervals) were generated by bootstrapping (resampling with replacement 1,000 times).

the most stable conformation in both in solution and when membrane bound would often remain in the neighborhood of the crystal structure reference. In the membrane-bound state, we have observed that the major molecular orientation states (labeled as "Xtal", "m1", and "m2") display considerable variation in their molecular tilt angles (Fig 3D). Specifically, the Xtal reference state features two density peaks at approximately 5- and 37-degrees tilt, whereas the m1 state is compactly centered around 24 degrees, and the m2 state exhibits a wide range of molecular tilt angles from 0 to approximately 68 degrees (Fig 3D).

To further elucidate the range of conformational changes exhibited by FVa during membrane binding in the simulations, we calculate the root-mean-squared deviation (RMSD) of its domains and the entire molecule, comparing them with the reference X-tal structure (S2 Fig).

The RMSD plots are shown to indicate that the intradomain structural deviations are negligible, except for the A3. The global alignment of the entire FVa (S2 Fig, top right panels) will tend to cover up much of what is going on. Instead, however, aligning the heavy chain readily allows tracking that it is C2 and not C1 that is mostly flexible relative to the whole FVa when they rearrange relative to each other. It's worth noting that the movement and flexibility observed around the most flexible hinge, the C1-C2 connection, also lead to alterations in the relationship between the C2 domain and the A1 domain, minimally affecting the inter-domain distance but severely affecting the domain-domain interface by altering the number of contacts made among A1 and C2 (S3 Fig).

### Certain amino acid residues exhibit contact preference toward interacting with PS lipids

In FVa, certain amino acid residues exhibit a preference for binding to phosphatidylserine (PS) lipids as opposed to phosphatidylcholine (PC) lipids. This preference is likely due to the electrostatic interactions that occur between the amino acid residues and the head group of the PS lipid, which has a net charge of -1e. By quantifying the frequency of contact between the amino acid residues and the two types of lipids in simulation models of membrane-bound FVa, we can gain insight into which specific residues of the protein are in contact with which lipids, and how often. In Fig 4A, the contact frequency for the C1 and C2 domains of the FVa protein is shown, with most trajectories exhibiting nonzero contact frequencies (except for Memb #1, which only makes minor and transient contacts in the A3 domain). The regions of the membrane with which the protein makes contact are generally confined to the spikes that insert into the membrane and adjacent areas, depending on the domain tilt. Comparing the interactions between the membrane and the C1 and C2 domains, we observe that C2 tends to make more frequent contact with PS than C1, but in either case, most of the protein-lipid contacts occur between the protein and PC lipids (Fig 4B, indicated by wider and longer-tailed PC density in the split violin plot of the average contact frequencies between the amino acid residues of the protein and the bilayer lipids). This is likely due to the stoichiometric ratio of the bilayer, which is PC:PS = 4:1. Finally, we note that no deep insertion of a significant portion of either discoidin domains is observed (Fig 4C).

To further explore the preference for PS over PC, we can compare the contact frequency for each amino acid residue in the FVa protein with PS versus PC. This analysis is shown in Fig 5, which highlights the specific residues that exhibit a strong preference for PS over PC. From a structural perspective, it becomes clear that these residues are located on the surface of FVa (Fig 5A). In particular, residues R1907, L1908, N1913, and R2027 in the C1 domain and K2087, K2157, and K2187 in the C2 domain exhibit the strongest preference for PS over PC (Fig 5B). To better visualize the point cloud for low contact frequencies, logarithmic scales are used to stretch the data (Fig 5A, bottom).

### The spikes of the discoidin domains insert into the hydrophobic region of the lipid membrane

The discoidin domains of FVa are characterized by spikes that insert into the hydrophobic, or non-polar, region of the membrane. The quantitative extent of insertion of each discoidin domain can be determined by the height of the spike 1–3 amino acid residues in relation to the membrane interior ($z = 0$) when in the membrane-bound state (Fig 6). Notably, a trend is observed whereby the length of the loop correlates with the extent of insertion of spike residues located within that loop (Fig 6). However, this correlation may not hold universally among discoidin domains. In addition, it is observed that centrally located

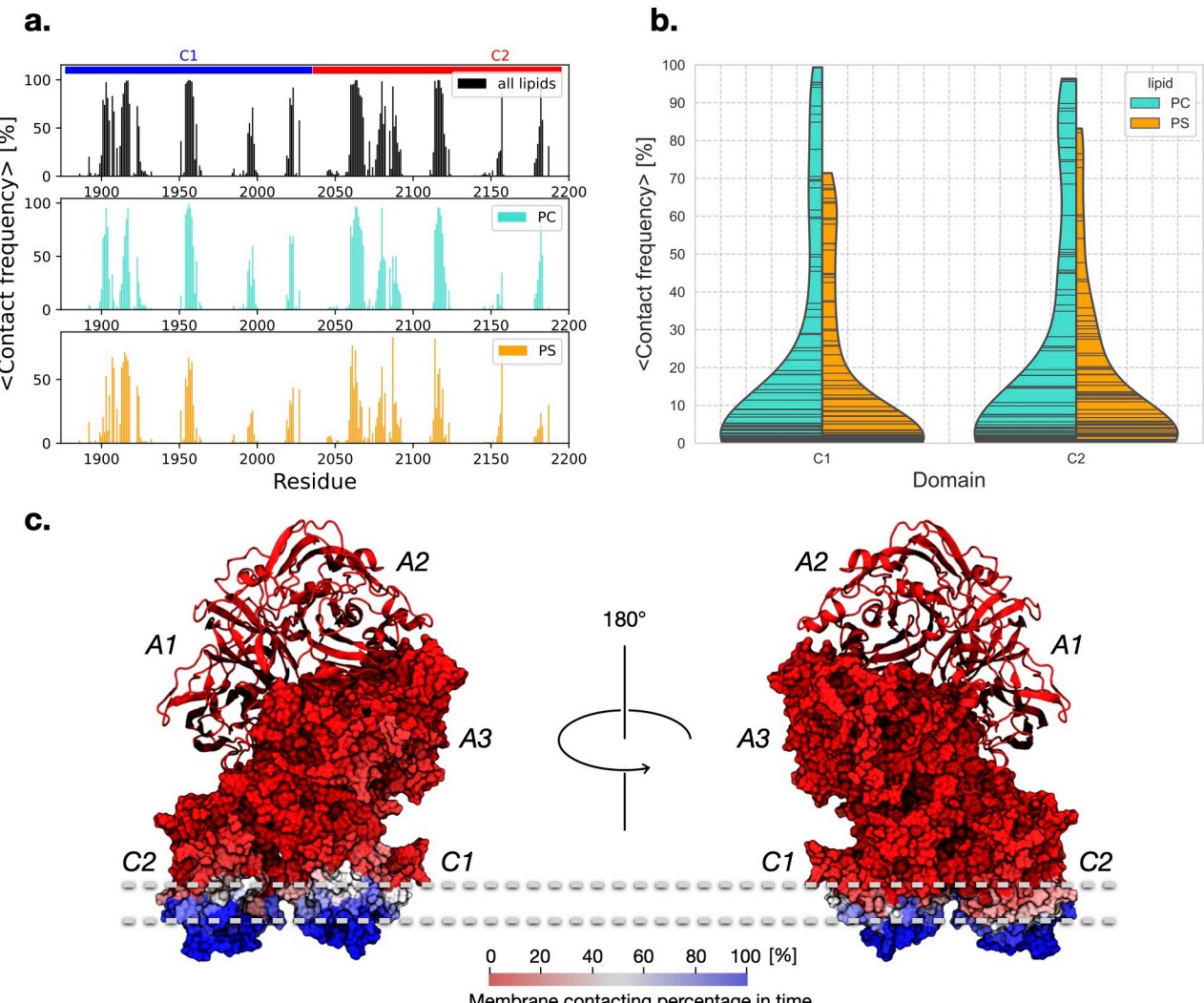

**Fig 4. a.** Quantification of average contact frequencies between the amino acid residues of the protein and the lipids of the bilayer. **b.** The split violin plot shows the average contact frequencies between the amino acid residues of the protein and the bilayer lipids, with the left-hand side indicating contacts with PC (in turquoise) and the right-hand side indicating contacts with PS (in orange), offering insights into the variability of contacts within each domain-lipid combination. The contacts are separated over the two discoidin domains, C1 and C2, with the width of each violin indicating data density. The "stick" inner plot shows individual data points. **c.** Mapping of the contact frequencies to the structure of FVa, where the heavy chain is shown as a ribbon diagram and the light chain is shown as a surface plot. The color scale ranges from red (0% contact) to blue (100% contact), with white representing intermediate values (~50% contact).

hydrophobic residue(s) enable deeper penetration for C2 spikes 1 and 3 (W2063 and W2064 for spike 1, L2116 for spike 3), whereas R2080 exhibits the deepest insertion for the C2 spike 2. This ability of charged spike residues to insert into the bilayer, rather than solely anchoring on top, allows the hydrophobic residues of the spikes to embed into the hydrophobic core of the bilayer more deeply and thereby anchor the domain. C1 spike 1 is truncated by 5 amino acids and hardly inserts below the level of the average P atom position. C1 spikes 2 and 3 roughly equal their counterparts in C2 in terms insertion with C1 spike 2 slightly shallower and C1 spike 3 slightly deeper on the average without differences being statistically significant (Fig 6)

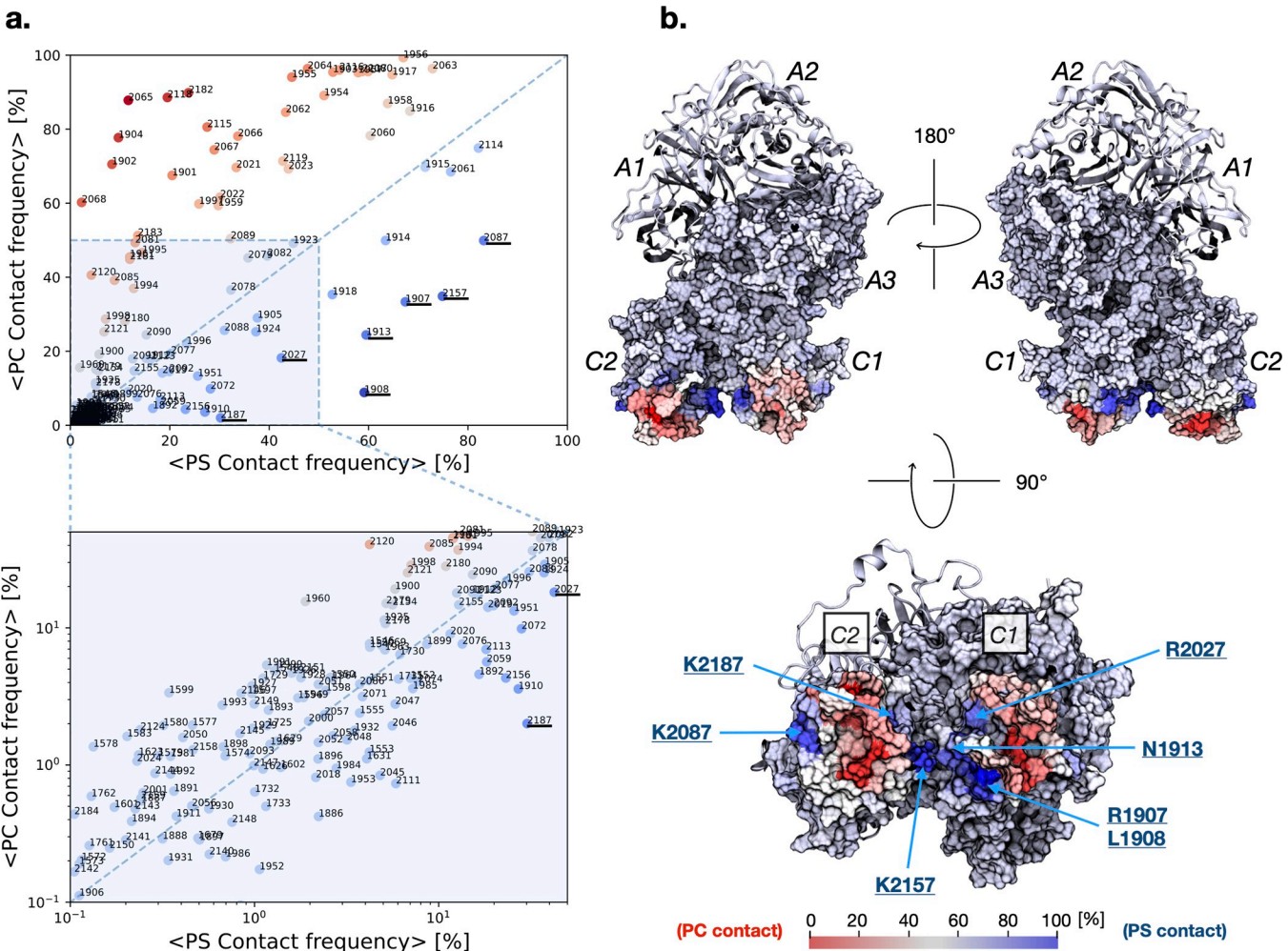

**Fig 5.** **a.** Amino acid residues of FVa that may contribute to its specificity towards phosphatidylserine (PS) lipids. The candidates were identified based on the average frequency of contact between the residues and PS and phosphatidylcholine (PC) lipids observed in the simulations. Each data point is color-coded according to its deviation from the diagonal line, which represents equal contact frequency to PS and PC. Blue points indicate a preference for PS, while red points indicate a preference for PC. The zoomed-in region (*bottom*) is shown using double logarithmic scale to stretch the data point cloud. **b.** The contact frequency data from panel **a** is mapped onto the surface structure of FVa, with the heavy chain represented as a ribbon diagram and the light chain as a surface plot. The color scale is the same as in panel **a**, ranging from red (preference for PC) to blue (preference for PS), with white indicating no preference (i.e., equal contact frequency to both lipids).

## Discussion

The specific effects described in the research article related to the binding and interaction of FVa with membranes may be somewhat unique to FVa or other proteins with similar domain architecture and structural features. However, similar effects are likely relevant to other proteins containing discoidin-type domains that bind to and interact with membranes. One example is the blood coagulation protein FVIIIa, which also contains dual discoidin domains and has been shown to interact with and be stabilized by phospholipid membranes [33,47,48]. FVIIIa has been measured by FRET to interact with a phospholipid surface at a highly tilted molecular orientation [39], similar to the extreme case that we observe here for FVa in one of the trajectories (Fig 2, Memb #1).

More broadly, the binding process, in which sidechains of basic residues (that are located above the binding spikes) first make contact with PS headgroups as if probing and selecting

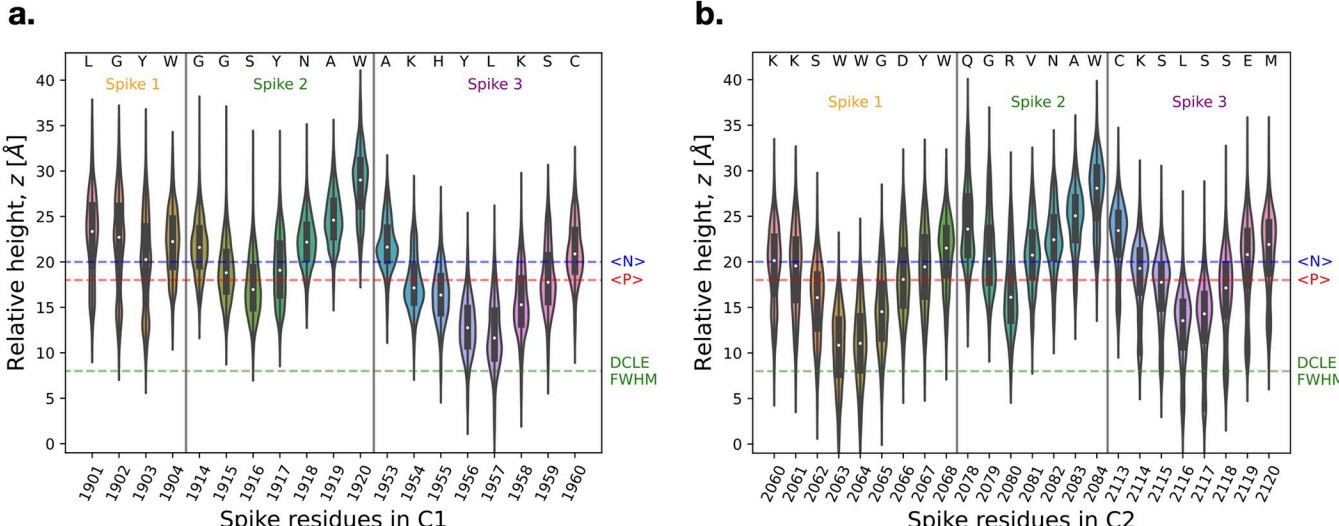

**Fig 6. Membrane spikes insertion.** The height of amino acid residues in the C1 (panel **a**) and C2 (panel **b**) spikes 1–3 of FVa is shown relative to the membrane center ($z = 0$), pooled over the membrane-bound portion of the trajectories (t = 100–500 ns). The distribution of relative heights is visualized using violin plots, with the width representing the kernel density estimate. The black box indicates the 95% confidence interval, and the white dot indicates the median of the data. The average positions of lipid headgroup nitrogen atoms, <N> (in blue), and phosphorous atoms, <P> (in red), are indicated by dashed lines. "FWHM" = Full Width at Half Maximum of the DCLE density histogram along the membrane normal direction, $z$.

the binding spot, followed by membrane insertion of the hydrophobic spikes, actually seems to be a common strategy of peripheral membrane proteins with hydrophobic loops (or hydrophobic protrusions) [49]. In addition, it has been suggested (by MD) that the FYVE domain of early endosomal antigen (EEA1) first makes contact with phosphatidylinositol-3-phosphate (PI(3)P) in the membrane and subsequently inserts its loops into the membrane [50]. Furthermore, there are MD/modeling studies of the PX domain on the membrane [51,52], in which β1-β2 loop is inserted into the membrane in advance of establishing PX domain-PI(3)P binding. In the whole membrane binding process, however, a PI(3)P may first start contacting with the PX domain, preceding the β1-β2 loop insertion. As for the PH domain, a membrane-bound model for the PH domain of PLEKHA7 has been proposed [53], in which hydrophobic loops and basic residues outside or the perimeters of the loops are in contact with the membrane. PH domains may therefore also conduct membrane binding in the "basic residues first, hydrophobic loops afterwards" manner.

## Dynamic pirouette-like motion

Our results are in support of a "pirouette" or "carousel" mechanism for membrane binding of FVa, which is a conceptual model that describes how the coordinated binding of C1 and C2 domains allows for dynamic movement of the two domains, enabling them to interact with different regions of the membrane in a flexible and adaptable manner that also depends in the molecular orientation of whole protein molecule. For example, Ref. [48] proposed an initial membrane-binding of FVIIIa C2 through a rotating motion of the protein around a central axis, which allowed C1 to interact with different membrane regions in a flexible and dynamic manner [48]. However, it is important to note that the study in Ref. [48] used a coarse-grained model known to speed up dynamics [54,55] and looked only on the minimal construct FVIIIa C1-C2. While the FVIIIa or FVa light chain (A3-C1-C2) is commonly used in experimental studies of membrane binding [56–58], there is limited experimental information available on a C1-C2 construct from either FVIIIa or FVa. Nevertheless, it is certainly expected that the

C1-C2 construct exhibits greater conformational flexibility than the light chain, which in turn is more flexible than the full-length FVIIIa/FVa. Seemingly, there is little to no difference in membrane-binding among the unactivated FV and FVa. Ref. [59] conducted a study on the surface binding of the FVa light chain, including binding to phospholipid membranes. Their findings agree with the observations made in Ref. [48] and with our own results [33]. The idea of a dynamic pirouette-like motion is an interesting and still developing concept in the study of membrane-protein interactions. Lastly, we remark that significant structural deviations have been measured experimentally as well using AFM [60].

While the specific effects described in this work may be unique to FVa and its structural cousin FVIIIa, similar binding and interaction mechanisms have been observed in other proteins that interact with and are stabilized by membranes. For example, neuropilin-1 [61], discoidin domain receptor proteins [62], and lactadherin [63], all containing the discoidin-type domain, have been shown to interact with membranes, although it remains to be explored whether all these proteins will bind the membrane in the exact same fashion. Interestingly, FVIIIa C1, FVIIIa C2, FVa C2, and lactadherin C2 have been studied using computational methods, and the similarities between their membrane-binding mechanisms on the domain anchor are striking [33,48,59,64–70].

## "Anything But Choline" (ABC) hypothesis for FVa

The ABC hypothesis is the well-established observation that for GLA domains, there are two different types of binding interactions between the GLA-bound calcium ions and membrane lipids: one is PS specific, in which both carboxy and phosphate groups of a PS headgroup interact with a calcium ion; the other is only phosphate group of any phosphate-including headgroup except PC interacts with a calcium ion, bending over the rest of its headgroup [71].

While the binding of FVa to the phospholipid membrane is calcium-independent, a similar notion has been proposed for FVa based on the apparent requirement for charged lipids (typically PS).[71] To our knowledge, and based on our current study, we do not find compelling evidence of an ABC hypothesis for FVa and the molecular basis of lipid specificity for FVa (or FVIIIa) remains not well understood. For instance, the spike sequences of the C2 domain are similar but not identical in FV and FVIII. Surprisingly, swapping the spikes incurs changes in lipid specificity of membrane binding and also biological cofactor activity [72,73]. To further explore the possible connection to the ABC hypothesis idea and to reveal the molecular basis lipid specificity in various mixtures, simulations with membranes of many different lipid compositions are needed.

## Possibility of membrane allostery

Allosteric regulation is commonly found in biological processes. In the coagulation cascade, numerous allosteric effectors have been characterized, usually enzyme cofactors, ions, or phospholipids necessary to enhance the activity of proteases and ramp up the response [74–86]. The traditional view is that the membrane's role in this picture is to increase the effective concentration by confining the binding partners to a two-dimensional plane and aligning them properly. Our simulation further indicates that the membrane may also have a role in determining the conformations of FVa that can bind productively to its target protease, factor Xa, by selecting for specific conformations of FVa to bind the membrane. In other words, the membrane as a whole can possibly be considered as an allosteric effector of coagulation proteins, which is a novel and intriguing concept. This expanded view emphasizes the multifaceted nature of allosteric regulation and underscores the importance of investigating the molecular details of the membrane-proteins interplay.

## Comments on the need to modify the HMMM model

The highly mobile membrane-mimetic (HMMM) model was first used to simulate membrane binding of the GLA domain of human coagulation factor VIIa (FVIIa) [31] and since then it has been employed for many membrane proteins [32–34]. The HMMM model has been considered as one of most efficient settings when included in membrane systems to simulate in order to study membrane binding mechanisms of membrane proteins, particularly peripheral membrane proteins, and their complex formation on the membrane at atomic scale [87–89]. The HMMM system of the GLA domain [31] was not very large (~0.07M atoms with a ~90x90Å$^2$ membrane patch), and minor problems unnoticeable in such smaller systems may become evident when the HMMM model is employed for larger membrane systems, such as those in the current study (~0.3M atoms with a ~140x140Å$^2$ membrane). One such inconvenience is short-tailed lipids (st-lipids) popping out into bulk water. As shown in Fig 2 of Ref. [31] (at ~15–20 ns), st-lipids are mobile not only laterally but also vertically. St-lipids' popping-out occasionally happen and they often come back to the water/lipid interface in a short time. For systems with a larger membrane patch, the popping-out would occur more often and the more st-lipids may stay in bulk water. To reduce such popping-out, one would consider a simple-minded, makeshift solution of imposing harmonic potentials along the membrane normal (along the $z$-axis when the membrane patch spans over an x-y plane) on a certain atom in st-lipids. Such potentials do work out to eliminate the pop-ups, but unnatural; the legitimate choice of harmonic constants and the reference points for not causing artifacts cannot be readily sought. We instead chose little longer acyl tails in order to make st-lipids at the water/DCLE interface favors the DCLE layer more and pack more orderly at the interface. Another inconvenience is undulation of the membrane patch on the absence of restraints except for the periodic boundary conditions (PBC). Possible configuration of the membrane patch in a *small* system is significantly reduced due to PBC from the actual/corresponding system in nature (and therefore self-formation of the HMMM membrane is quick as shown in Fig 2 of Ref. [31]). For systems with larger size of a membrane patch, the restriction by PBC becomes less significant. As a result, the membrane patch tends to undulate, which may be more significant for thinner, and loose (as consist of smaller molecules) membrane patch like the HMMM. In order to minimize the undulations, we applied well-shape potentials to DCLE molecules so that they will mostly stay within a certain slab (*i.e.*, a certain range of $z$-coordinates). Note that the potentials are applied to no st-lipids but to DCLEs only, and only when DCLEs move outside the slab; st-lipids are free to diffuse around, and essentially no slowdown of MD rate by applying the well-shape potentials was seen.

The new settings of the HMMM model, which we termed "extHMMM", worked out with no noticeable inconvenience. We would suggest the extHMMM model as the new standard HMMM settings for generic use.

## Materials and methods

### Protein model and initial setup

We built the structural model of human coagulation FVa (corresponding to residues 29–2224 of UniProt entry P12259), following the procedures by Ref. [29]. This model contains the correct coordinated ions, post-translational modifications, and certain missing loops not resolved in the crystal structure models, making it the most complete structure available of human FVa. Three systems were prepared: two in which FVa with a membrane patch with different areas par lipid (APLs) and FVa in aqueous solution (Table 1). For the membrane system, the protein model is initialized above the membrane with its principal axis aligned with the membrane

**Table 1. Overview of three different sets of FVa simulations carried out and presented in this study (with two different membrane patches and in solution).**

| Trajectory tag | FVa model | extHMMM settings | | | | # atoms [x $10^5$] | Sim. Time [ns] | Data |
|---|---|---|---|---|---|---|---|---|
| | | Lipid mix PC:PS | Lipid tail | rAPL [%] | Restraints | | | |
| Memb #1 | Ref. [29] | 4:1 | dioctanoyl | 110 | DCLE well | 3.12 | 500 | Ref. [97] |
| Memb #2 | Ref. [29] | 4:1 | dioctanoyl | 110 | DCLE well | 3.12 | 500 | Ref. [97] |
| Memb #3 | Ref. [29] | 4:1 | dioctanoyl | 110 | DCLE well | 3.12 | 500 | Ref. [98] |
| Memb #4 | Ref. [29] | 4:1 | dioctanoyl | 120 | DCLE well | 3.14 | 500 | Ref. [98] |
| Memb #5 | Ref. [29] | 4:1 | dioctanoyl | 120 | DCLE well | 3.14 | 500 | Ref. [99] |
| Memb #6 | Ref. [29] | 4:1 | dioctanoyl | 120 | DCLE well | 3.14 | 500 | Ref. [99] |
| Soln #1 | Ref. [29] | N/A | N/A | N/A | N/A | 2.98 | 400 | Ref. [100] |
| Soln #2 | Ref. [29] | N/A | N/A | N/A | N/A | 2.98 | 400 | Ref. [100] |
| Soln #3 | Ref. [29] | N/A | N/A | N/A | N/A | 2.98 | 400 | Ref. [100] |

"N/A" indicates that a particular quantity is not applicable to the corresponding simulation. "rAPL" = Relative area per lipid.

normal and discoidin domains facing the membrane at an initial distance of ~1 nm between the membrane surface and the closest α-Carbon so that only a few atoms of FVa have near-zero electrostatic energies with membrane lipids and the vast majority of FVa atoms have no interaction with the membrane. This initial positioning guides C1 and C2 to bind the membrane, consistent with experiment. Thus, some flexibility is maintained while reducing initial configuration-dependency of the membrane binding processes within the experimentally established framework. Two relative APLs (rAPLs), 110% and 120% of the experimentally obtained APLs [44–46] to test any influence over the efficiency and the converged states of membrane binding of FVa.

## Bilayer lipid membrane model: extHMMM, Application of HMMM to systems of extended size require modifying both lipid topology and restraint

The HMMM membrane patch consisted of medium-length-tailed phospholipids (either phosphatidylcholine or phosphatidylserine, in a stoichiometric ratio of 4:1 in both leaflets) sandwiching a layer of a layer of 1,1-dichloroethane (DCLE) solvent molecules with the lipid headgroups facing the water phase and acyl chains against the DCLE organic phase. We employed dioctanoylphosphatidylcholine (diocta-PC, 8:0/8:0) and dioctanoylphosphatidylserine (diocta-PS, 8:0/8:0) lipids, whose acyl tail length is increased by three as compared with the original divalerylphosphatidylcholine (DVPC, 5:0/5:0) and divalerylphosphatidylserine (DVPS, 5:0/5:0). The membrane patch was stabilized by a flatbottom positional restraint on DCLE molecules along the membrane normal (z-axis) according to the equations

$$u = k*dr^2$$

$$dr = \max(0, r - rfb)$$

$$r = \text{abs}(z - z0)$$

with $z0 = 0$, $rfb = 1.05$ nm, and $k = 10$ kJ mol$^{-1}$ nm$^{-2}$. By utilizing both the restraint scheme and the longer lipid tails, the HMMM patch was effectively stabilized, thereby preventing DCLE leakage and lipids from popping out, all without imposing any restraints on other components of the system. We refer to these new settings of the HMMM model as "extHMMM" since it

has lipids with extended acyl tails and improved restraints for general use of the HMMM model in larger-scale systems. Membrane lipid were arranged at 110 and 120% rAPL relative to the experimentally obtained standard APL for POPC and POPS of 68.3 and 60.4 Å$^2$ / lipid, respectively [44–46], with the CHARMM-GUI builder [87,90].

## MD simulations

The MD systems consisting of a HMMM patch and the protein were solvated with TIP3P water [91] and NaCl added (to a physiological concentration of 150 mM) using the CHARMM-GUI builder [87,90]. Reference system was prepared in a similar fashion and contained only FVa in bulk water at a physiological NaCl concentration of 150 mM. The details of the individual systems simulated are listed in Table 1. The standard CHARMM-GUI protocol for minimization and relaxation were employed. Production MD simulations were performed as three independent runs for each system (Table 1) in the *NPnAT* ensemble (for two membrane systems; *Pn* refers to the pressure control by the size control along the membrane normal, i.e., the *z*-axis, sustaining the membrane patch dimensions, i.e., the x- and y-lengths, constant) and *NPT* ensemble (for the solution system) with the target pressure and temperature of 1 atm and 310 K, controlled by the Monte Carlo barostat [92] (coupling frequency of 100 steps) and the Langevin thermostat [93] (friction coefficient of 1 ps$^{-1}$), respectively. Long-range electrostatic forces were calculated using the particle mesh Ewald (PME) method with a grid spacing of approximately 1 Å and 5th-order B-splines for interpolation [94]. Electrostatic forces were switched starting at 10 Å and switched entirely off by 12 Å. Periodic boundary conditions were applied in the *x*-, *y*-, and *z*-directions. All MD simulations were performed using OpenMM 7 [95] with the CHARMM36m.[96] force field. An integration time step of 2.0 fs was used for the velocity Verlet algorithm. The lengths of all bonds that involve a hydrogen atom were constrained. Simulation trajectories are deposited at Zenodo [97–100]. The computation rate was about 15 ns/day for the membrane systems (approx. 3.12 x 10$^5$ atoms) on a server equipped with two Intel Xeon Silver 4214 CPUs (each consisting of 24 of 2.20 GHz cores) and the NVIDIA RTX 2080 Ti GPU.

## Analysis and visualization

Analyses and visualization of molecular structures and trajectories were conducted using VMD 1.9.1 [101], PyMOL (version 2.5.0, Schrödinger LLC) and in-house scripts. Scripts in python used for analysis and plotting relied on packages MDAnalysis [102], NumPy [103], pandas [104], Matplotlib [105], and seaborn [106]. For quantification of residue-residue or residue-lipid contacts, a cut-off distance of 5.0 Å was used. The tilt angle, α, is defined as the angle between the FVa first principal axis (proxied by the vector from the center of mass of C1-C2, residues 1879–2033 + 2038–2193, α-Carbons only, to the center of mass of A2, residues 320–655, α-Carbons only) and the membrane normal (*z*-axis). The φ (C1-C2 "scissors") is defined as the angle spanned by the C1 domain center of mass (residues 1879–2033, α-Carbons only), the C1-C2 linker (residue 2037, α-Carbon), and the C2 domain center of mass (residues 2038–2193, α-Carbons only). Similarly, the θ (C2 out-of-plane "kick") angle is defined the dihedral spanned by the C2 first principal axis (proxied by α-Carbons of residues 2186 and 2127) and the C1 first principal axis (proxied by α-Carbons of residues 1967 and 2026).

## Supporting information

**S1 Fig. Visualization of the FVa molecular surface.** (**a**) Representation with color-coding based on residue type, showing the exposed protein backbone (in white), positive residues (in

blue), negative residues (in red), polar residues (in green), and hydrophobic residues (in orange). The arrow indicates the position of C2's spike 1 residues W2063 and W2064. (**b**) Representation showing the electrostatic surface of the FVa molecule, computed using PyMOL's vacuum electrostatics functionality. The emphasized regions highlight the location of the "charges belt" of positive residues on C1 and C2.
(TIFF)

**S2 Fig. Root-mean-square deviation (RMSD) of Cα positions, a measure of distance between corresponding atoms in two structures, for the different domains of FVa in the the membrane-containing simulations (Memb #1-#6) based on various superimposed selections.** The different panels include RMSD calculation for multiple superimposed selections each corresponding to a specific superimposed alignment applied before calculating the RMSD, enabling comparison between both the different simulations and domains of FVa. The term "si" indicates which selection was superimposed before RMSD calculation.
(TIFF)

**S3 Fig. FVa conformational variability.** The number of contact points (Cα distance within 10.0 Å) between residues of the A1 and C2 domains as well as the A1 and C2 center of mass (c. o.m.) inter-domain distance in the membrane-containing simulations (Memb #1-#6), both measures of the dynamic interactions between these domains during the simulations.
(TIFF)

## Acknowledgments

J.J.M. gratefully acknowledges the research framework provided by Research Computing at University of South Florida. All simulations were conducted at the advanced computing resources at the University of South Florida.

## Author Contributions

**Conceptualization:** Jesper J. Madsen, Y. Zenmei Ohkubo.

**Data curation:** Jesper J. Madsen.

**Formal analysis:** Jesper J. Madsen.

**Investigation:** Jesper J. Madsen.

**Methodology:** Jesper J. Madsen, Y. Zenmei Ohkubo.

**Visualization:** Jesper J. Madsen, Y. Zenmei Ohkubo.

**Writing – original draft:** Jesper J. Madsen, Y. Zenmei Ohkubo.

**Writing – review & editing:** Jesper J. Madsen, Y. Zenmei Ohkubo.

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
