## [Decision Letter · Decision Letter 0]

31 Oct 2023

Dear Dr. Madsen,

Thank you very much for submitting your manuscript "Elucidating the complex membrane binding of a protein with multiple anchoring domains" for consideration at PLOS Computational Biology.

As with all papers reviewed by the journal, your manuscript was reviewed by members of the editorial board and by several independent reviewers. In light of the reviews (below this email), we would like to invite the resubmission of a significantly-revised version that takes into account the reviewers' comments.

In particular the second reviewer raises a number of important points both about simulation setups and conclusions. We would particularly like you to adress the issues raised about the influence of the single (?) starting conformation; more quantitative justification for the conclusions would strengthen the results (like a potential-of-mean-force), or e.g. showing that starting with other domains facing the membrane does not lead to binding.  The significant deviation of the domain conformations from the crystal structure is also something that potentially gives rise for concern, and we would prefer to see some sort of analysis or argument (based e.g. on other studies) why this is not a simulation artefact.

We cannot make any decision about publication until we have seen the revised manuscript and your response to the reviewers' comments. Your revised manuscript is also likely to be sent to reviewers for further evaluation.

Sincerely,

Erik Lindahl

Guest Editor

PLOS Computational Biology

Nir Ben-Tal

Section Editor

PLOS Computational Biology

Reviewer's Responses to Questions

**Comments to the Authors:**

Reviewer #1: Uploaded as an attachment.

Reviewer #2: Madsen and Ohkubo's research investigates the interaction between the Coagulation factor Va (FVa) protein and a membrane composed of PC and PS (in a 4:1 ratio) through molecular dynamics simulations. They employed a modified version of the highly mobile membrane-mimetic (HMMM) model to achieve this. In this revised model, they extended the length of the lipid chains from 5 to 8 carbons (dioctanoyl). This extension was crucial to creating a sufficiently long hydrophobic core that could accurately replicate the interactions between FVa and lipids, especially when FVa is deeply embedded within the membrane. This innovative approach is clever and significant.

Their study addresses five key aspects:

i) Investigating the binding motif mechanism of FVa's multiple anchoring domains.

ii) Analyzing the tilting behavior of the protein when it binds to the membrane.

iii) Comparing the structural disparities between the crystallographic structure and the model in both the membrane and aqueous environments.

iv) Characterizing the interactions between lipids and the protein.

This research represents a noteworthy contribution to understanding the complex interactions between FVa and lipid membranes, shedding light on important aspects of this biological process.

However, I have several concerns regarding the construction of the systems, which may not be conducive to a comprehensive study of the binding mechanism (point (i) ). In particular, how the systems were built seems inadequate for addressing the mode of FVa membrane binding. Systems were constructed with the C2 domain facing the membrane at a distance of 1nm, while the cutoff values for coulombic and van der Waals interactions are set at 1.2. Consequently, these components are not at a non-interacting distance as one would expect. Therefore, the author's claim that "Initial contact and insertion of the C2 and C1 domain spikes progress sequentially" is not surprising, given the bias in their system. Biased initial conditions influence the initial systems.

The system is large, but the author could have improved the study's validity by commencing with multiple random orientations where the C2 and C1 domains do not face the membrane. This would have allowed for a more thorough exploration of the binding motif. If the author intends to delve deeper into which of the two domains initiates the binding process, the authors should consider employing umbrella sampling to generate a free energy profile. This additional approach would provide a more comprehensive understanding of the energetic landscape governing the binding of these domains to the membrane. It's important to note that the chosen approach is valid only when the binding motif is already known.

Given the simulations' limitations, it is impossible to derive any insights about the binding mechanism, specifically whether it involves C2 or C1 first. Consequently, any statements regarding the binding mechanism (C2 vs C1) should be omitted from the discussion to maintain scientific accuracy.

Major comments:

1) Furthermore, the methodology employed for constructing the initial structures needs more clarity. Did the author generate random initial velocities, or were six separate membrane constructions carried out? I could only infer this information by downloading and inspecting the trajectory as a reader. To enhance transparency and reproducibility, such crucial details should be explicitly outlined in the "Materials and Methods" section of the study. Additionally, it would be advantageous to include a concise summary of this information in the Results section, not solely confined to the Methods section.

2) The results presented in Figure S1 and S2 require more comprehensive explanations. The authors mention what has been calculated, but what specific findings have been derived from the data needs to be clarified. The figure displays RMSD values without effectively highlighting the key results or insights obtained. Additional context and interpretation of the figure's data would enhance its clarity and usefulness. Furthermore, I couldn't discern any notable differences in the A1-C2 distance. Maybe, it would have been beneficial to generate a two-dimensional histogram encompassing all trajectories to address this issue. This histogram could have depicted the relationship between the center of mass (COM) distance and the number of contacts, shedding light on whether a correlation exists between a high number of contacts and a shorter distance (and vice versa). This is just a suggestion; I want to mention you can display the result differently.

3) The absence of substantial differences between PS and PC contacts in Figure 4a is noteworthy. The authors attribute this observation to the stoichiometric ratio of the bilayer, PC: PS=4:1. While this hypothesis appears plausible, it can be supported by calculating the contact frequency while normalizing it against the total number of lipids.

Furthermore, there is a need for clarity in explaining how the frequency of contact is calculated and subsequently normalized. This crucial procedure needs more specification in the figure caption and/or the Material and Methods section.

Additionally, it is crucial to address whether any portions of the trajectory have been excluded from the analysis, as this information is presently only indicated in Figure 6. Furthermore, error bars are missing.

4) In addition to the contact frequency analysis, exploring the interaction energies, encompassing both van der Waals (vdW) and Coulombic interactions could be highly beneficial between PC and PS. Such an investigation will likely reveal substantial disparities in these interactions, offering further insights into the nature of the interactions.

5) Figure 6 exhibits high distribution of data, making it challenging to draw conclusive findings. The issue may stem from including excessive frames and incorporating all trajectories. To address this concern effectively, the author should consider classifying the orientations, possibly based on tilting, and subsequently analyze and average frames with similar orientations. This approach will likely enhance the results' reliability and provide a more robust basis for conclusions. Same approach can be used in aother analysis too.

6) Regarding the use of two different rAPLs, the study does not explain, nor does it discuss the unique characteristics, benefits, drawbacks, or potential effects on the results. It remains unclear whether specific features, such as tilting, contacts, and binding motifs, are influenced by rAPL. Averaging all trajectories together might not be the right choice since systems might behave differently.

7) I noticed an unusual structure in the Discussion section. The authors are more focused on justifying the importance of their study by comparing it to other proteins and mechanisms rather than delving into interpreting their results. It gives the impression of a review-style format with multiple paragraphs, each starting with an introduction. Instead, a more concise approach is advisable. The primary focus should be on discussing their work, with a brief concluding section addressing the potential relevance to other proteins.

Similarly, the Comments section on modifying the HMMM model could be integrated into the main discussion without requiring a separate header. Openly discussing these points within the context of the broader discussion would create a more cohesive and streamlined narrative.

Minor comments:

- In the Material and Methods section, there is mention of an "NPnAT ensemble." Is it a typo?

- In Figure 4d, it would be beneficial to explicitly specify which section of the membrane is represented by the dotted lines for clarity.

- The majority of the references conclude with the notations "From NLM PubMed-not-MEDLINE" and "From NLM Medline." Consider removing it.

Reviewer #3: In this paper the authors have discussed the complexities of protein-lipid interactions in the context of FVa. They use a new molecular dynamics method termed “extHMMM” to identify key mechanisms of FVa binding to the membrane such as molecular tilt, domain organization, and contact preferences.

The reviewer recommends the article be published after minor revision. The reviewer has the following comments:

1. Abstract

There are two typos in the abstract.

2. Introduction

The “extHMMM” model is very important for the basis of this study, the reviewer suggests adding more background on the model in the introduction paragraph.

3. Results: Mode of FVa membrane binding

The reviewer suggests further discussion of the following topics: What are the further differences between the binding patterns of FVa to membrane surfaces and those observed in lipid-decorated nanotube models, and what implications does this have for understanding how FVa interacts with regular cell membranes functionally?

4. Results: Domain organization of FVa significantly deviates from the crystallographic reference structure both when membrane-bound state and in solution state, with deviation being more pronounced in the solution state

The reviewer suggests further discussion of the following topics: In the context of FVa's membrane binding how do the observed variations in the φ and θ angles of FVa in solution state and membrane-bound state relate to its potential functional significance?

5. Results: Certain amino acid residues exhibit contact preference toward interacting with PS lipids

The reviewer suggests further discussion of the following topics: How does discoidin domain absence affect FVa's membrane-bound conformation and interaction with lipid bilayers? How might this lack of deep insertion influence the functional role of FVa in membrane binding? Is there a time-scale difference between first contact with PS lipids over PC lipids before complete insertion?

6. Results: The spikes of the discoidin domains insert into the hydrophobic region of the lipid membrane

The reviewer suggests further discussion of the following topics: What is the effect of the depth of insertion of spike residues (e.g., W2063, W2064, L2116, R2080) into the hydrophobic core of the bilayer on anchoring and stability of discoidin domains? Are there functional implications associated with varying insertion depths for different spikes? Do these spikes show changes in structures upon insertion?

7. Discussion

Could you elaborate on the implications of the membrane potentially acting as an allosteric effector of FVa? Are there experimental studies that support the concept of membrane allostery in coagulation proteins?

**Have the authors made all data and (if applicable) computational code underlying the findings in their manuscript fully available?**

Reviewer #1: **No: **Source code for analysis is not included.

Reviewer #2: Yes

Reviewer #3: Yes

PLOS authors have the option to publish the peer review history of their article (what does this mean?). If published, this will include your full peer review and any attached files.

Reviewer #1: No

Reviewer #2: No

Reviewer #3: No

Figure Files:

While revising your submission, please upload your figure files to the Preflight Analysis and Conver

---

## [Decision Letter · Decision Letter 1]

4 Apr 2024

Dear Dr. Madsen,

Thank you very much for submitting your manuscript "Elucidating the complex membrane binding of a protein with multiple anchoring domains using extHMMM" for consideration at PLOS Computational Biology. As with all papers reviewed by the journal, your manuscript was reviewed by members of the editorial board and by several independent reviewers.

Both reviewers are in general happy with your revised version, and we will be delighted to accept it for publication, but we think the additional comment by reviewer #2 is very valid, so we would ask you to modify your methods description based on the reviewer's suggestion. We also recommend that the analysis script be made available openly. But leave it for you to decide.

Sincerely,

Erik Lindahl

Guest Editor

PLOS Computational Biology

Nir Ben-Tal

Section Editor

PLOS Computational Biology

Reviewer's Responses to Questions

**Comments to the Authors:**

Reviewer #1: All but one of my concerns is addressed. I would strongly recommend to open source the analysis scripts. While they may be 'straightforward applications', it is impossible to quickly reproduce the results without them, and making the code available on request adds an unnecessary barrier to accessing the code, and increases the chance that at some point in the future it is lost.

Reviewer #2: The authors have addressed my questions and comments to my satisfaction overall. However, a significant issue I raised still needs to be addressed.

The authors introduced the following new text in the Materials and Methods section, which I found incorrect.

"For the membrane system, the protein model is initialized above the membrane with its principal axis aligned with the membrane normal and discoidin

domains facing the membrane at an initial distance of ~1 nm between the membrane surface and the closest α-Carbon so that only a few atoms of FVa have near-zero

electrostatic energies with membrane lipids and the vast majority of FVa atoms have no interaction with the membrane. This initial positioning minimizes initial configuration dependency of the membrane binding processes (if observed), while effectively trim irrelevant detours, given that experiments indicate C1 and C2 are membrane-binding domains".

Although the authors' initial setup successfully eliminates unnecessary deviations, it is inaccurate to claim that "This initial positioning minimizes initial configuration dependency of the membrane binding processes ". In reality, the methodology introduced by the authors imposes a significant bias on the membrane binding processes.

I agree with the authors' reasoning for their chosen methodology, recognizing it as logical and well-founded. However, their analysis cannot be used to clarify how the protein's binding motif transitions from an aqueous environment to the membrane. Consequently, I suggest revisions to the section titled "Mode of FVa Membrane Binding."

I suggest relocating this section to the Materials and Methods or the Supplementary Information to detail the construction of the membrane-bound state utilized in subsequent analyses. The rationale for this suggestion stems from the observation that the claims within this section lack statistical support and merely describe visual observations from the simulations. Providing a more precise explanation of the methodology in a more appropriate section would enhance the clarity of the subsequent research findings.

**Have the authors made all data and (if applicable) computational code underlying the findings in their manuscript fully available?**

Reviewer #1: **No: **See comments to authors (analysis scripts).

Reviewer #2: Yes

PLOS authors have the option to publish the peer review history of their article (what does this mean?). If published, this will include your full peer review and any attached files.

Reviewer #1: No

Reviewer #2: No

Figure Files:

Data Requirements:

Reproducibility:

References:

---

## [Decision Letter · Decision Letter 2]

19 Jun 2024

Dear Dr. Madsen,

We are pleased to inform you that your manuscript 'Elucidating the complex membrane binding of a protein with multiple anchoring domains using extHMMM' has been provisionally accepted for publication in PLOS Computational Biology.

Best regards,

Erik Lindahl

Guest Editor

PLOS Computational Biology

Nir Ben-Tal

Section Editor

PLOS Computational Biology

Reviewer's Responses to Questions

**Comments to the Authors:**

Reviewer #2: I am satisfied with the author responses.

**Have the authors made all data and (if applicable) computational code underlying the findings in their manuscript fully available?**

Reviewer #2: None

PLOS authors have the option to publish the peer review history of their article (what does this mean?). If published, this will include your full peer review and any attached files.

Reviewer #2: No

---

## [Editor Report · Acceptance letter]

1 Jul 2024

PCOMPBIOL-D-23-01274R2 

Elucidating the complex membrane binding of a protein with multiple anchoring domains using extHMMM

Dear Dr Madsen,

I am pleased to inform you that your manuscript has been formally accepted for publication in PLOS Computational Biology. Your manuscript is now with our production department and you will be notified of the publication date in due course.

With kind regards,

Jazmin Toth
